# Integrated Biological and Chemical Control against the Maize Late Wilt Agent *Magnaporthiopsis maydis*

Asaf Gordani [1,2], Bayan Hijazi [2], Elhanan Dimant [1] and Ofir Degani [1,2,*]

[1] Plant Sciences Department, Migal—Galilee Research Institute, Tarshish 2, Kiryat Shmona 1101600, Israel
[2] Faculty of Sciences, Tel-Hai College, Upper Galilee, Tel-Hai 1220800, Israel
\* Correspondence: d-ofir@migal.org.il or ofird@telhai.ac.il; Tel.: +972-54-678-0114

**Abstract:** Today's fungal plant disease control efforts tend towards environmentally friendly and reduced chemical applications. While traditional broad-spectrum fungicides provide efficient protection to many field crops, they pose a risk to the soil's beneficial microflora and a potential health hazard. Moreover, their intensive use often evokes the appearance of resistant pathogens. On the other hand, biocontrol agents such as *Trichoderma* spp. provide a green solution but often cannot shield the plants from aggressive disease outbreaks. Integrated biological and chemical disease control can combine the benefits of both methods while reducing their drawbacks. In the current study, such a bio-chemo approach was developed and evaluated for the first time against the maize late wilt pathogen, *Magnaporthiopsis maydis*. Combinations of four *Trichoderma* species and Azoxystrobin were tested, starting with an in vitro seed assay, then a growth room sprouts trial, and finally a semi-field, full-season pot experiment. In the plates assay, all four *Trichoderma* species, *Trichoderma sp.* O.Y. (T14707), *T. longibrachiatum* (T7407), *T. asperellum* (P1) and *T. asperelloides* (T203), grew (but with some delay) in the presence of Azoxystrobin minimal inhibition concentration (0.005 mg/L). The latter two species provided high protection to sprouts in the growth room and to potted plants throughout a full season in a semi-field open-enclosure trial. At harvest, the P1 and T203 bio-shielding exhibited the best parameters (statistically significant) in plant growth promotion, yield increase and late wilt protection (up to 29% health recovery and 94% pathogen suppression tracked by real-time PCR). When applied alone, the Azoxystrobin treatment provided minor (insignificant) protection. Adding this fungicide to *Trichoderma* spp. resulted in similar (statistically equal) results to their sole application. Still, the fact that Azoxystrobin is harmless to the beneficial *Trichoderma* species over a complete semi-field condition is a great opening stage for carrying out follow-up studies validating the integrated control in a commercial field situation challenged with acute disease stress.

**Keywords:** Azoxystrobin; *Cephalosporium maydis*; crop protection; disease control; fungus; *Harpophora maydis*; late wilt; *Magnaporthiopsis maydis*; maize; *Trichoderma*



## 1. Introduction

The phytopathogenic fungus *Magnaporthiopsis maydis* (previously known as *Cephalosporium maydis* or *Harpophora maydis* [1–3]) is a major threat to commercial maize production in highly impacted areas such as Egypt [4,5], Israel [6], Spain [7], Portugal [8] and India [9]. The resulting disease is typified by the rapid wilting of plants towards the end of the season [10]. This alarming situation results from water transport blocking by the pathogen, which gradually worsens from days 50–60 of growth, when the male flowering starts [11–13]. The pathogen, which is soil- [14] and seed-borne [15], thrives both on leaving tissues or dead plants (and is thus considered a hemibiotroph), spreads itself using fruiting bodies (spores and sclerotia) [16,17] or survives on alternative host plants [18]. Reduced host specialization might occur since *M. maydis* may become hostile to cotton plants in certain cases [19].

The affected maize crops suffer from severe dehydration at the final disease stages and critical (50–100%) yield loss occurs [20–23]. The disease incidences depend on local condi-

tions, pathogen population and soil infestation degree [18]. Drought stress is recognized as a major reducing factor for a plant's ability to cope with the disease [24]. Indeed, evidence shows that soil moisture is one of the most crucial factors in late wilt disease outbursts and severity [7,24–27].

Today, control methods rely on host resistance, i.e., developing maize hybrid lines that show late wilt disease (LWD) (immunity (latest studies are [5,6,9,28–33]). Still, such a method is not without flaws. If a market-preferable maize genotype is also highly resistant, it will be cultivated existentially and may eventually lead to its resistance breakdown [6]. The reason for this unstable resistance is the presence of a highly virulent *M. maydis* strain in the pathogen population [12,34–37], and the pathogen's ability to colonize susceptible cultivars asymptomatically (or with minor symptoms) and manifest itself through their seeds [15,16]. Thus, other control methods were tested over the years (summarized in [38]). Among many compounds tested against the late wilt pathogen (for example, [39–41]), Azoxystrobin (Amistar S.C.; Syngenta, Basel, Switzerland, supplied by Adama Makhteshim, Ashdod, Israel) stood out in several studies carried out against the Israeli *M. maydis* population. A trial series proved that applying this compound in three 15-day (from sowing) intervals using drip line irrigation in proximity to the plants (to avoid loss of material in the ground) can protect the plants from LWD even in severe cases [22,23,38].

However, drip line irrigation is considered expensive and unrealistic in many commercial cornfields. Azoxystrobin seed coating is a cheap method tailored to any cultivation method. Still, it can only provide protection up to a certain point through an additional protective layer, which could make a difference in mild disease outbreaks [22]. Moreover, the world's scientific effort is leaning towards seeking environmentally friendly solutions that involve minimal use of chemo-pesticides. Today, phytopthologists are trying to identify new biocontrol agents in plants and soil.

Biological methods based on bacteria and fungi originating from the soil [42], seeds [43], and even from surprising sources such as marine sponges [44], have proven themselves to protect against growth suppression and prevent wilt symptoms caused by *M. maydis*. One example is the bacteria *Bacillus subtilis* and *Pseudomonas* spp. [45,46]. In a field experiment, a mixture of *B. subtilis* and *Pseudomonas koreensis* greatly reduced *M. maydis* infection and increased yield parameters [46]. It also increased the thickness of the layer surrounding the maize stem vascular bundles (sclerenchymatous sheath). Another well-studied research direction is using species of the genus *Trichoderma* as biocontrol agents. Such studies conducted in Egypt were based on *T. harzianum*, *T. koningii*, *T. viride* and *T. virens* [47–49]. In Israel, successful treatments were achieved utilizing *T. longibrachiatum*, *T. asperelloides* and *T. asperellum* [50,51]. Thus, why not use this biological approach exclusively? Because in severe disease cases, it may not be sufficient.

Although *Trichoderma*-based biological control is widely studied for its eco-protection advantage, its application is often challenged by natural stress in farming, leading to unpredictable control effects [52–54]. A proposed new control strategy maintains high chemical effectiveness while drastically reducing the dosages applied by combining both chemical and biological control methods [55]. It was previously reported that the synergistic application of low-toxic chemical fungicides and biocontrol agents could improve biocontrol stability and efficiency against plant diseases, ultimately reducing the use of chemical fungicides. An example is the combined application of *Trichoderma harzianum* (strain SH2303) and difenoconazole-propiconazole in controlling maize southern corn leaf blight disease caused by *Cochliobolus heterostrophus* [56]. Such an environmentally friendly solution might also prevent the increasing fungicide resistance problem (reducing the selection pressure on pathogens caused by pesticide overuse, thereby, the chances of resistance evolving).

Yet, as far as we know, such an approach has never been tested against the maize late wilt pathogen. In the current work, this method was tested using the already field-validated Azoxystrobin compound [22,23,38] and the three *Trichoderma* species that had gained success in our previous studies, *T. longibrachiatum*, *T. asperelloides* and *T. asperellum* [50,51]. To achieve this purpose, we performed a series of experiments at an increasing complexity

level, including lab (colony growth and seeds) assays, growth-room sprouts pathogenicity assay and a semi-field full-season pot trial. The maize growth parameters evaluation and disease symptoms estimation were accompanied by quantitative real-time PCR (qPCR) tracking of *M. maydis* DNA inside the host tissues.

## 2. Materials and Methods

### 2.1. Rationale of the Study Design

To reduce Azoxystrobin to a minimum, we ran a plate assay (see Section 2.3) with different preparation concentrations and set the minimal inhibition concentration (MIC) point needed to control the maize late wilt (LWD) pathogen, *Magnaporthiopsis maydis*. This concentration was used in another plate assay to identify the possible sensitivity to this fungicide of the selected protective *Trichoderma* species. The final lab stage before the growth experiments aimed at verifying that the *Trichoderma* species have no inhibition impact on seed germination and the sprouts' initial development. It was unnecessary to test the seeds' sensitivity to possible Azoxystrobin phytotoxicity since no such effect was identified in previous studies [22,57,58]. The growth experiments were conducted in two steps: first, the integrated chem-bio control method was evaluated in a growing room under controlled conditions in sprouts (up to 20 days); second, the protocol was tested in a semi-field, open-enclosure full-season pots trial.

### 2.2. Fungal Source and Growth Conditions

One representative isolate of *M. maydis* called *Hm-2* (CBS 133165) was chosen for this study. This isolate was deposited in the Fungal Biodiversity Center, CBS-KNAW, Utrecht, The Netherlands, after being characterized by its virulence towards maize, physiology, colony morphology, and microscopic and molecular traits [16]. It was formerly recovered in 2001 from a Kibbutz Sde Nehemia cornfield in the Hula Valley (Upper Galilee, northern Israel) from Jubilee cv. maize plants that showed dehydration symptoms. It was recently evaluated and compared to 15 other *M. maydis* strains in our fungal library collected from maize fields across Israel, and proved to be a mild aggressive—virulence rank 7 out of 17 strains [35]. *M. maydis* was grown in a solid potato dextrose agar (PDA; Difco Laboratories, Detroit, MI, USA) medium by transferring 6-mm-diameter disks from the edges of a young culture that had grown for 4–6 days to a new growth plate. The plates were incubated at 28 ± 1 °C in the dark.

The *Trichoderma* species selected for this study (Table 1) were previously proven efficient in controlling LWD in mycoparasitism assay (Figure 1) and whole growth season trials [50,51,59]—*T. longibrachiatum* (T7404), *T. asperelloides* (T203) and *T. asperellum* (P1). An additional species, *Trichoderma* sp. O.Y. 14707 (T14707 [44]), had no antagonistic ability against *M. maydis* in a plate mycoparasitism assay [50]. In the plate confront test, the T14707 colony was almost completely covered by *M. maydis* mycelium (Figure 1). Thus, this *Trichoderma* isolate was chosen for the current study to provide a negative control. The lab growth conditions for the *Trichoderma* species were identical to those of *M. maydis* growth described above. For the growth of mycelium in a liquid medium, five *Trichoderma* sp. colony discs were inserted into an Erlenmeyer flask with 150 mL of potato dextrose broth (PDB; Difco Laboratories, Detroit, MI, USA). The cultures were incubated for 10 days in a shaking incubator at 150 rpm at 28 ± 1 °C in the dark. The liquid growth medium was filtered using a Buchner funnel with double Whatman paper, centrifuged at 6000 rpm for 20 min and further filtered through a 0.4-micron membrane. The sterilized growth fluid was used in the seed assay described below.

**Table 1.** List of *Trichoderma* spp. isolates used in this study.

| Species | Designation | Origin | Reference | Role |
|---|---|---|---|---|
| *Trichoderma asperelloides* | T203 | ATCC 36042, CBS 396.92 | [50,60] | *M. maydis* antagonist |
| *Trichoderma asperellum* | P1 | *Zea mays*, Prelude cv. | [43] | *M. maydis* antagonist |
| *Trichoderma longibrachiatum* | T7407 | *Psammocinia* sp. [1] | [44,50] | *M. maydis* antagonist |
| *Trichoderma* sp. O.Y. 14707 | T14707 | *Psammocinia* sp. [1] | [44] | Control (non-influencing) |

[1] Mediterranean sponge *Psammocinia* sp.

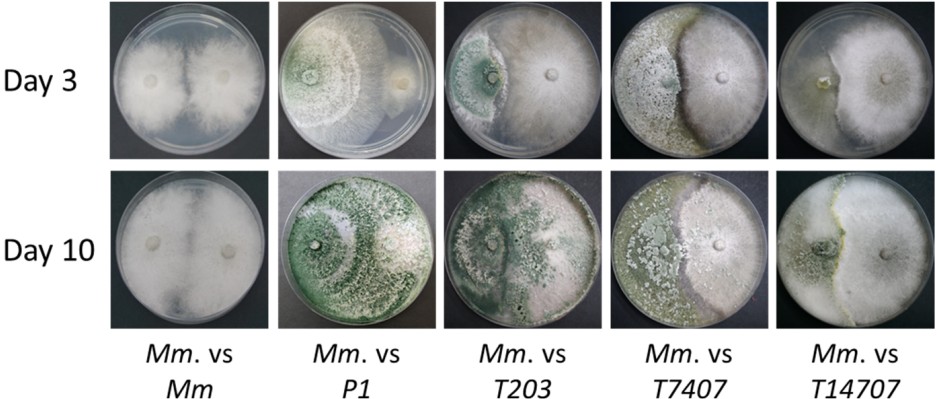

**Figure 1.** Mycoparasitism of *Trichoderma* spp. against the maize late wilt pathogen *Magnaporthiopsis maydis*. The photos were adapted from [18,50] with modifications. *Trichoderma* isolates (shown in Table 1) were seeded on the left and *M. maydis* (*Mm*) on the right. The *Trichoderma* species that restrict the pathogen's growth or that grow above the pathogen's mycelium are considered to have mycoparasitic potential. As can be seen, the control (non-influencing) *Trichoderma* sp. O.Y. 14707 was covered almost entirely by the *M. maydis* mycelia. The fungi were sown on rich solid substrate (PDA) plates and incubated for 3 and 10 days at 28 ± 1 °C in the dark (except for P1, which was grown for 3 and 6 days). The control is a growth plate where *M. maydis* was seeded in both poles (*Mm* vs. *Mm*). Each photo is representative of five independent replicates for each confrontation assay.

### 2.3. Azoxystrobin Inhibition Plates Assay

For the Azoxystrobin (Sigma-Aldrich, Rehovot, Israel) inhibition plates assay, the compound was added to the PDA at a final concentration of 0–0.016 mg/L active ingredient per plate to challenge *M. maydis* or 0–10 mg/L active ingredient per plate to test its influence on the *Trichoderma* species (Table 1). Control was the same medium plate without the fungicide. The assay was performed in 5–7 replications. Each plate was seeded with a 6-mm-diameter *M. maydis* or *Trichoderma* sp. mycelial disc (cut from the boundary of a fresh colony grown as described in Section 2.2). The plates were incubated at 28 ± 1 °C in the dark. The colonies' growth radius was measured every two days on two perpendicular planes.

### 2.4. In Vitro Seed Assay

The sweet corn Prelude (SRS Snowy River Seeds, Orbost, Australia, provided by Green 2000 Ltd., Bitan Aharon, Israel) and Megaton (Limagrain, Saint-Beauzire, France, marketed by Hazera Seeds Ltd.) cultivars were selected for this study. Both are highly susceptible to late wilt, as previously mentioned [59]. The seeds of these cultivars were rinsed in 1% sodium hypochlorite (NaOCl) and sterilized DDW. They were then submerged in *Trichoderma* spp. growth media for six hours. The control seeds were submerged in sterile tap water. The seeds were then dried, placed in Petri dishes with sterilized tap water-soaked Whatman paper and incubated at 28 ± 1 °C for a week in the dark. Each group of seeds was tested in six repetitions. Each replication is a plate containing 10 seeds. Germination percentages were tested every day for up to six days. The sprout's biomass

and epicotyl length were measured at the experiment's end. The experiment was performed in 5–6 biological repetitions.

### 2.5. Growth Room Sprouts Assay

#### 2.5.1. Preparation of Infected Sterilized Millet Grains

Infected sterilized millet (*Panicum miliaceum*) kernels were used to spread *M. maydis* in the soil, as previously described [59]. Millet grains (1.5 kg) were kept for half an hour in boiling tap water and then transferred to a 2 L glass jar. The millet grains were mixed with 19.5 g of gypsum powder (CaSO$_4$·2H$_2$O) to raise the pH and sterilized by autoclave for 0.5 h at a temperature of 120 °C. An 8-day-old colony of *M. maydis* (whole PDA culture with the gel from a 9-cm-diameter Petri plate) was added to the jar, and the grains were mixed with the fungus with a sterilized spatula. The jar was closed with a loose seal that allowed for gas exchange and aluminum foil and incubated at 28 ± 1 °C in the dark for 14 days until the mycelium covered the grains.

#### 2.5.2. *Magnaporthiopsis Maydis* Infection and Protective Treatments

The soil was inoculated one week before the experiment by adding 8 g sterilized *M. maydis*-infected millet grains (see Section 2.5.1) to the top layer of the pot's top 5 cm soil. Complementary infection was done on the sowing day by adding three *M. maydis* colony agar discs (see Section 2.2) to each maize seed. The non-infected control pots were left untreated.

Four *Trichoderma* isolates (Table 1) were selected for the pot experiment based on their performance in an in vitro assay (dual culture confront test, Figure 1), in sprouts and in complete season growth trials [50,51,59]. Three of them, T203 (*T. asperelloides*), P1 (*T. asperellum*) and T7407 (*T. longibrachiatum*), excelled in these tests, exhibiting high biocontrol behavior against the LWD pathogen. The fourth, T14707, lacked any antagonistic activity against *M. maydis* and was chosen as a reference strain and a control. The *Trichoderma*-based protection treatments were conducted in two steps: first, the maize seeds were submerged in the *Trichoderma* spp. growth media for 10 min (see Section 2.4); second, each maize seed was enriched by three *Trichoderma* spp. colony agar discs (see Section 2.2) with the sowing. The chemical treatment was done by adding 0.25 mL of the Azoxystrobin preparation (Amistar S.C., which includes 0.0625 mg of the active ingredient) to each pot with the irrigation. This procedure was applied once, 10 days after sowing.

#### 2.5.3. The Growth Room Assay Protocol

Pots (2.5-L) were filled with 30% Perlite No. 4 (for aeration) and 70% heavy local soil from the farm where the whole-season pots assay was conducted (see Section 2.6.1 below). The same ground was used in the two experiments, the growth room sprouts assay and the farm full season trial. Osmocote (ScottsMiracle-Gro, Marysville, OH, USA), a six-month-release fertilizer, was added to each pot according to the manufacturer's instructions. The Prelude maize cv. seeds (after being soaked for 10 min in the *Trichoderma* spp. growth media as described above or in distilled water in the control grains) were planted in the pots ca. 5 cm beneath the soil surface. Every pot was seeded with five seeds. The pots were maintained in a growth room under a photoperiod of 16 h light and 8 h dark. The growth room conditions were 45% humidity and 27 ± 2 °C. Immediately after sowing, the pots were irrigated to initiate germination. Irrigation was carried out by a computerized system at 40 mL once every two days. During the test, fertilization treatments and treatments against various pests were performed according to the Israel Ministry of Agriculture Consultation Service (SAHAM). The experiment was conducted in 9–10 repetitions per treatment (each repetition is a pot containing five sprouts).

At the end of the experiment (day 20), all seedlings were uprooted, thoroughly cleaned with running tap water and dried using paper towels. All plants were subjected to wet biomass, shoot length and phenological stage (number of leaves) estimation. In addition, root samples (0.7-g) were taken from the plants of each pot for DNA extraction.

### 2.6. Whole-Season Pots Assay under Semi-Field Conditions

2.6.1. Growth Protocol and Conditions

In this test carried out at the North R&D plantation farm (Hula Valley, Upper Galilee, northern Israel, 33°09′08.2″ N 35°37′21.6″ E), the four *Trichoderma* species (Table 1), alone or in combination with Azoxystrobin, were tested against unprotected control plants in *M. maydis*-enriched soil. The assay was carried out in 9–10 repetitions for the treatment (each repetition was a pot containing five plants). Pots were filled with 10 L of heavy local soil from the farm having no history of LWD. If such an infestation occurred, it was expected to be minor. The soil was mixed with 30% coarse Perlite for aeration. Computerized irrigation was carried out with drip lines once a day, with the amount adjusted to maintain moderate humidity conditions (usually 1.3 L per pot/day). Throughout the season, fertilizers and treatments against various pests were applied to keep the disease factor only from the *M. maydis* infection.

2.6.2. *Magnaporthiopsis Maydis* Infection and Protective Treatments

Inoculation was performed by mixing the upper part of the soil (the pot's top 7 cm soil) with sterilized millet grains (150 g prepared as described in Section 2.5.1). This procedure took place one week before sowing. A complementary soil infection was conducted with sowing and 9 days afterward by adding three *M. maydis* colony mycelia discs (see Section 2.2) to each maize seed. The control pots were kept pathogen-free. The biological and chemical treatments were conducted similarly to the growth room sprouts experiment (Section 2.5.2). Three *Trichoderma* spp. colony agar discs were added to each seed with the sowing. The Azoxystrobin (5 mL of the Amistar S.C., a preparation that includes 1.25 mg of the active ingredient per pot) was applied at three intervals from the seeding: after 15, 30 and 44 days. This chemical preparation concentration (0.5 mL per kg soil) was four times higher than the concentration of the growth room trial (0.0125 mL per kg soil). We considered the biomass of the plants (up to maturity versus sprouts in the growing room) and the Azoxystrobin movement in the ground.

2.6.3. Important Dates and Meteorological Data

The bio-chemo control semi-field experiment was conducted in the summer of 2022. The key dates are detailed in Table 2. The temperatures and humidity parameters during the maize growing season were typical and were well suited to the disease development, according to [51]. The meteorological data are presented in Table 3.

**Table 2.** The semi-field pot experiment's dates.

| Date | Inoculation, Planting and Sprouting Assessment | Days from Sowing |
|---|---|---|
| 19/07/2022 | 1st inoculation (sterilized infected millet grains) | −7 |
| 02/08/2022 | Sowing and 2nd inoculation (three discs/seed) | 0 |
| 09/08/2022 | Aboveground sprouting | 7 |
| 11/08/2022 | 3rd inoculation (three discs/sprout) and soil surface peek evaluation | 9 |
| **Pesticide treatments** | | |
| 17/08/2022 | Pesticide I | 15 |
| 01/09/2022 | Pesticide II (15 days from Pesticide I) | 30 |
| 15/09/2022 | Pesticide III (14 days from Pesticide II) | 44 |
| **Sampling and harvest** | | |
| 11/09/2022 | Midseason sampling and thinning | 40 |
| 21/09/2022 | Male flowering peak | 50 |
| 01/10/2022 | End of fertilization period growth estimation | 61 |
| 19/10/2022 | Harvest and final sampling | 78 |

**Table 3.** Meteorological data for the 2022 semi-field experiments [1].

| Parameters | Value |
|---|---|
| Dates | 02/08/2022–19/10/2022 |
| Temperature (°C) | $26.6 \pm 5.5$ |
| Humidity (%) | $61.9 \pm 16.9$ |
| Soil temp. top 5 cm (°C) | $30.4 \pm 4.1$ |
| Radiation (W/m$^2$) | 238.3 |
| Precipitation (mm) | 4.6 |
| Evaporation (mm) | 457.7 |

[1] Average data ($\pm$standard deviation) according to the Israel Northern Research and Development, Hava 1 Meteorological Station.

### 2.6.4. Growth and Disease Estimation

After nine days, peek percentages above the soil surface were checked in each treatment. On days 40 (end of the sprouting phase), 61 (the V7-V8 plants' growth stage at the end of the fertilization stage—R1) and 78 (in the milk ripening stage—R5), the appearance of symptoms, the phenological phase, and the plants' wet weight and height were measured. During the season, the percentage of wilting leaves was tracked. At the end of the season, the total health status of the plants was evaluated based on four categories: 1—healthy, 2—symptoms, 3—diseased and 4—dead. Additionally, the presence of fungus DNA in the roots of each plant was quantified at growth days 40 and 78 by quantitative real-time PCR (qPCR).

### 2.7. Molecular Real-Time PCR Diagnostics

### 2.7.1. DNA Extraction

The plants' roots were washed thoroughly with running tap water then twice with sterile double-distilled water (DDW) and sliced into ca. 2 cm sections. The total weight of each repeat was adjusted to 0.7 g. The pathogen's DNA was isolated and extracted according to a previously published protocol [61] with slight modifications [57].

### 2.7.2. Real-Time PCR Technique

The qPCR method was performed using Applied Biosystems, Foster City, CA, USA, ABI-7900HT device (384 well plates). The technique relies on a standard qPCR protocol, which detects mRNA (cDNA), but is optimized to detect *M. maydis* DNA [22,59]. The qPCR reaction is performed in a total volume of 5 μL per reaction: 0.25 μL of each primer (Forward/Backward at a concentration of 10 μM), 2.5 μL of a ready reaction mixture—iTaq™ Universal SYBR Green Supermix solution (Bio-Rad Laboratories Ltd., Hercules, CA, USA) mix and 2 μL DNA template. Reaction conditions: 95 °C for 60 s, 40 cycles of 95 °C for 15 s, 60 °C for 30 s, and finally, creating a melting curve. The primers used are detailed in Table 4. The A200 primers amplify the segment-specific to *M. maydis*. The COX gene (codes for the enzyme cytochrome oxidase, the last enzyme in the cellular respiratory chain in the mitochondria) primers amplify a housekeeping gene to normalize the amount of *M. maydis* DNA according to the $\Delta$Ct model [62,63], and the same efficacy was assumed for all samples. All amplifications were performed in four repetitions.

**Table 4.** Primers for *Magnaporthiopsis maydis* detection [1].

| Pairs | Primer | Sequence | Uses | Amplification | References |
|---|---|---|---|---|---|
| Pair 1 | A200a-for<br>A200a-rev | 5′-CCGACGCCTAAAATACAGGA-3′<br>5′-GGGCTTTTTAGGGCCTTTTT-3′ | Target gene | 200 bp *M. maydis* species-specific fragment, qPCR cycling—27 or above | [16] |
| Pair 3 | COX-F<br>COX-R | 5′-GTATGCCACGTCGCATTCCAGA-3′<br>5′-CAACTACGGATATATAAGRRCCRR AACTG-3′ | Control | Cytochrome C oxidase (COX) gene product, qPCR cycling—27 or below | [64,65] |

[1] The R symbol represents Guanine or Adenine (purine). The synthesized primer contained a mixture of primers with both nucleotides.

### 2.8. Statistical Analysis

The plates in the lab experiments, the pots in the growth room and the open-enclosure trial were all scattered in a fully randomized design. All experiments were analyzed using the same statistical method performed with JMP software, version 15 (SAS Institute Inc., Cary, NC, USA). The differences in results were defined using the one-way analysis of variance (ANOVA) and a Student's t-test post hoc (without multiple tests correction, $p < 0.05$).

## 3. Results

This study examined, for the first time, an integrated control interphase against the maize late wilt disease (LWD) pathogen, *Magnaporthiopsis maydis*. This integrated control strategy is based on a combined Azoxystrobin-*Trichoderma* treatment. Three *Trichoderma* species were selected for this work based on their high performance in former studies: *T. longibrachiatum* (T7404), *T. asperelloides* (T203) and *T. asperellum* (P1, Table 1). The Azoxystrobin minimal inhibition concentration (MIC) was set, and it was verified that it could be combined well with the biological agents. The final evaluation was made in sprouts (20 days) and mature plants over an entire season (78 days).

### 3.1. Azoxystrobin Inhibition Plates Assay

In order to keep the Azoxystrobin as low as possible during the growth experiments, our first goal was to set its *M. maydis* minimal inhibition concentration (MIC). A plate assay done with increasing dosages of this preparation (up to 0.016 mg/L, Figure 2) showed that the inhibition curve stabilized, starting at a concentration of 0.008 mg/L. At this concentration, the inhibition rate was 39% on day 4 and 48% on day 6 ($p < 0.05$ compared to the control). Thus, a concentration of 0.0625 mg (per 2-L pot) of active ingredient was chosen for the subsequent growth room experiments. But before applying the pesticide to the plants, we verified that the above concentration would not be lethal to the *Trichoderma* biocontrol agents. So, an evaluation of Azoxystrobin in increasing concentrations (0.0001–10 mg/L) was tested against each of the *Trichoderma* species used (listed in Table 1). The result showed that the *Trichoderma* species can resist even high pesticide concentrations (Figure 3). At the relatively low fungicide concentration (0.005 mg/L), the *Trichoderma* spp. growth inhibition was 23–27% in the T14707 and P1 assay, and 42–56% in the T203 and T7407 assay ($p < 0.05$).

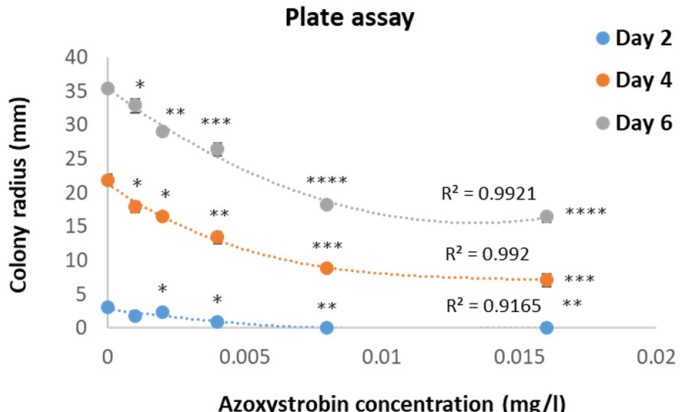

**Figure 2.** Azoxystrobin plate inhibition assay. The sensitivity of *Magnaporthiopsis maydis* to increased concentrations of the active ingredient (0–0.016 mg/L) was evaluated in solid potato dextrose agar (PDA) medium. Each growth plate was inoculated by a mycelial disk (6 mm in diameter) cut from the margin of 4–6-day-old colony. The cultures were incubated at $28 \pm 1$ °C in the dark for 2–6 days. Each treatment was conducted in 6–7 replications. Error bars represent standard error. Asterisks indicate a significant difference from the control without fungicide ($p < 0.05$ (*), $< 0.005$ (**), $< 0.0005$ (***), $< 0.00005$ (****)).

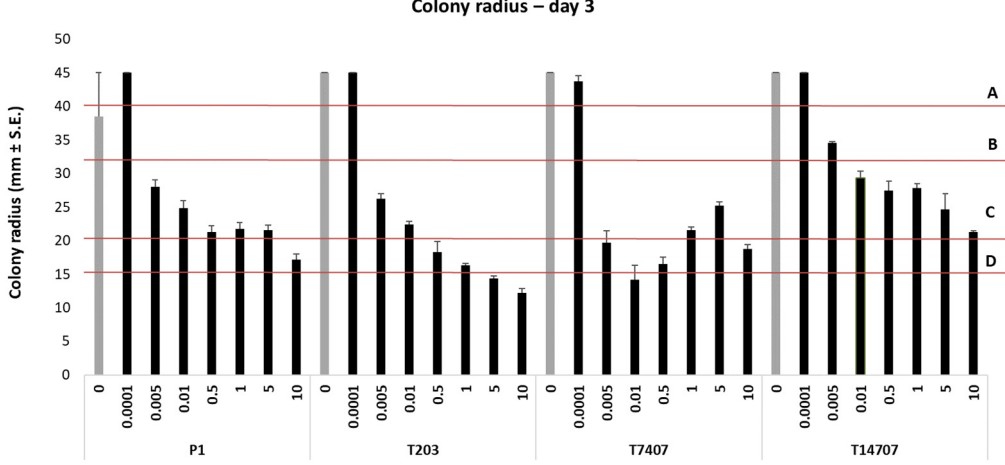

**Figure 3.** *Trichoderma* species Azoxystrobin sensitivity assay. The species are *T. asperellum* (P1), *T. asperelloides* (T203), *T. longibrachiatum* (T7404), and *Trichoderma* sp. O.Y. 14707 (T14707, see Table 1). The active ingredient (0–10 mg/L) was evaluated in PDA medium in 5–6 biological replications. Growth conditions are as in Figure 2. The colony radius was measured on day 3. Error bars indicate standard error. Different letters (A–D, on the right side of the chart) represent an ANOVA test significant difference ($p < 0.05$).

### 3.2. In Vitro Seed Assay

The seed germination and initial growth in the *Trichoderma* ssp.-secreted metabolites is an essential assay to ensure that no phytotoxic effect exists. If such an influence did occur, it should be mild, at the least. Indeed, such an inhibition impact (48%) was found to one of the bioprotective species, the T7407, in the Megaton maize cultivar (Figure 4A). Still, no such effect could be identified in the Prelude cultivar (Figure 4B). The fresh weight of the sprouts (Figure 4C) and epicotyl length (Figure 4D) after six days were similar in all *Trichoderma* treatments (but the Prelude cv. weight values were significantly higher). So, the Prelude cv. was chosen for the subsequent growth room and open-air trials.

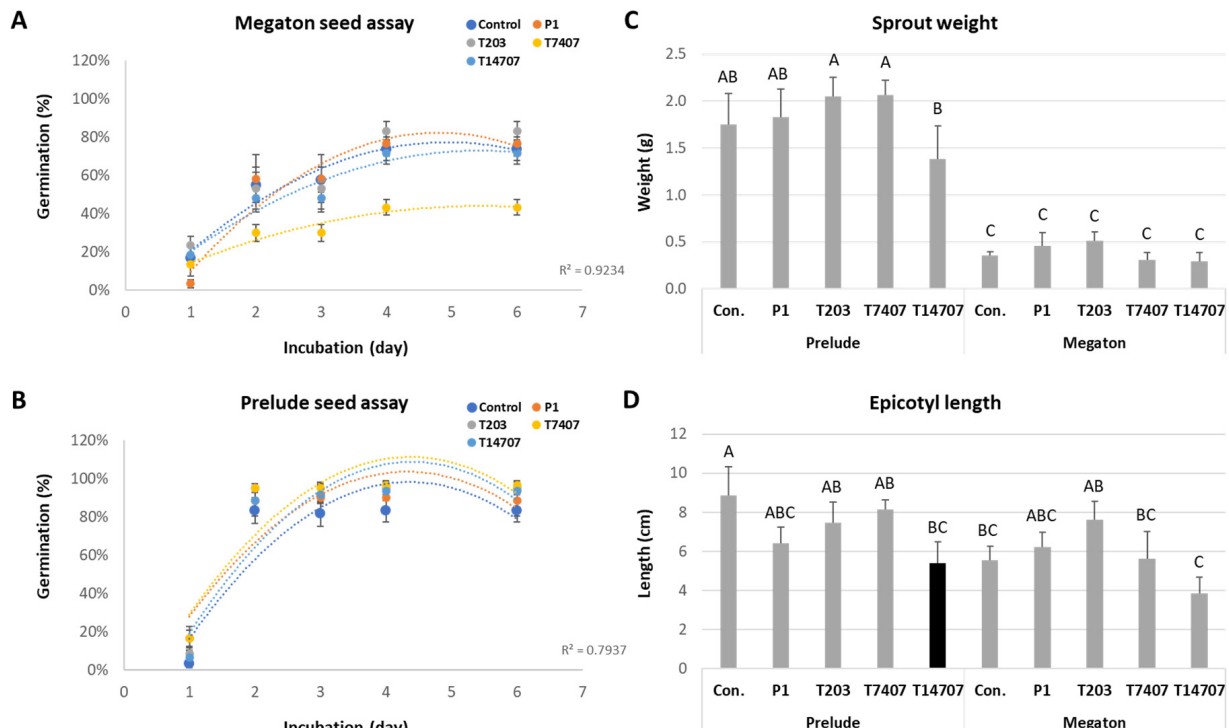

**Figure 4.** Seed assay for detecting the possible sensitivity to the biological treatments. The sweet corn Megaton (Limagrain, Saint-Beauzire, Puy-de-Dôme, France, marketed by Hazera Seeds Ltd.) and Prelude (SRS Snowy River Seeds, Australia, provided by Green 2000 Ltd., Bitan Aharon, Israel) cultivars were tested for susceptibility to *Trichoderma* treatments. The test was conducted in Petri dishes with wet Whatman paper at 28 ± 1 °C for one week in dark conditions. The seeds were dipped in the *Trichoderma* spp. (Table 1) growth media for six hours. Germination of the seeds of Megaton cv. (**A**) and Prelude cv. (**B**) was documented 0–6 days post-treatment (the only significant statistical difference from the control at $p < 0.05$ was found in the T7407 (*T. longibrachiatum*) treatment in the Megaton cv. test). The wet weight of the sprouts (**C**) and epicotyl length (**D**) were measured on day 6 of incubation. The experiment was performed in 5–6 biological repetitions. Error bars indicate standard error. Different letters (A–C) above the chart's bars represent an ANOVA test significant difference ($p < 0.05$).

### 3.3. Growth Room Sprouts Assay

Applying each biological agent in a growth room pots trial resulted in growth promotion after 20 days (Figure 5). This positive result was significant ($p < 0.05$) in the shoot weight (83%) and length (26%) of the *T. asperellum* (P1)-treated plants. Surprisingly, T14707 isolates that assumably shouldn't have any impact (included in the experiment as negative control) significantly excelled in this test and led to similar growth enhancement as P1 (80% and 23% shoot weight and length improvement). Another treatment, *T. asperelloides* (T203), produced a significant 23% plant height improvement and a 63% non-statistically significant fresh weight increase. While the Azoxystrobin sole treatment had a positive non-significant impact (for example, 45% shoot weight increase), its combination with the *Trichoderma* species greatly impacted the shoot parameters.

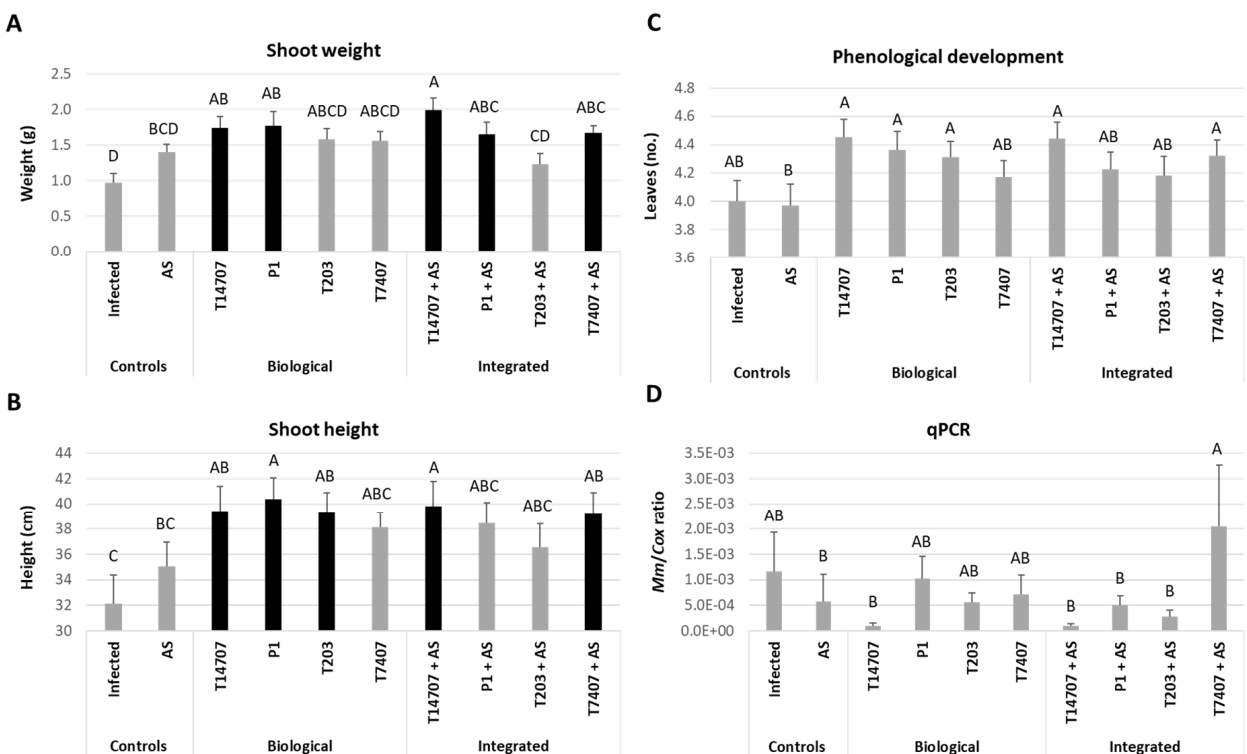

**Figure 5.** Growth room sprouts assay integrated control results. The Prelude maize cv. seeds were grown in *M. maydis* highly infected soil in 2-L pots for 20 days. While the control plants were left unprotected, chemical (0.25 mL/pot Azoxystrobin preparation (Amistar S.C.) with 0.0625 mg active ingredient), biological (*Trichoderma* spp., Table 1), or integrated bio-chemo shield treatments were applied. The evaluation included the shoot fresh weight of the plants (**A**), the shoot height (**B**), phenological development (number of leaves, (**C**)), and the relative amount of *M. maydis* DNA (*Mm*) normalized to the cytochrome C oxidase DNA (Cox) in the plants' roots (**D**). The experiment was conducted in 9–10 repetitions per treatment (each repetition is a pot containing five sprouts). Error bars indicate standard error. Different letters (A–D) above the chart's bars represent an ANOVA test significant difference ($p < 0.05$).

This integrated treatment performed best when the fungicide was combined with the T14707 strain (106% and 24% shoot weight and length improvement). Additionally, the bio-chemo application with *T. longibrachiatum* (T7404) and the P1 resulted in significantly induced aboveground growth in parts weight (72%) and height (23%) ($p < 0.05$). These results were also reflected in the plants' phenological development, but no statistical significance could be achieved. Except for the T7407, all combined treatments (Azoxystrobin with either T14707, P1, or T203) were better than the sole chemical or biological treatment. The T14707 treatment (with or without Azoxystrobin) resulted in a sharp 92% decrease in the pathogen's DNA compared to the non-treated control. P1 and T203 sole application led to 12% and 52% pathogen repression, respectively, but with Azoxystrobin, their impact increased drastically to 57% and 76%, respectively.

*3.4. Whole-Season Pots Assay under Semi-Field Conditions*

This open-enclosure full-season pots trial aimed at simulating field conditions as best as possible. At the same time, this type of growth enabled us to maintain control of the infection load, soil composition, watering regime and isolation of each treatment (while scattering the treatments in a fully randomized design, Figure 6A,B). As expected, the artificial soil inoculation succeeded in evoking intense disease outbreaks in the untreated control plants (Figure 6C–F). Indeed, the control plants suffered from 60% dehydration, as we will elaborate below.

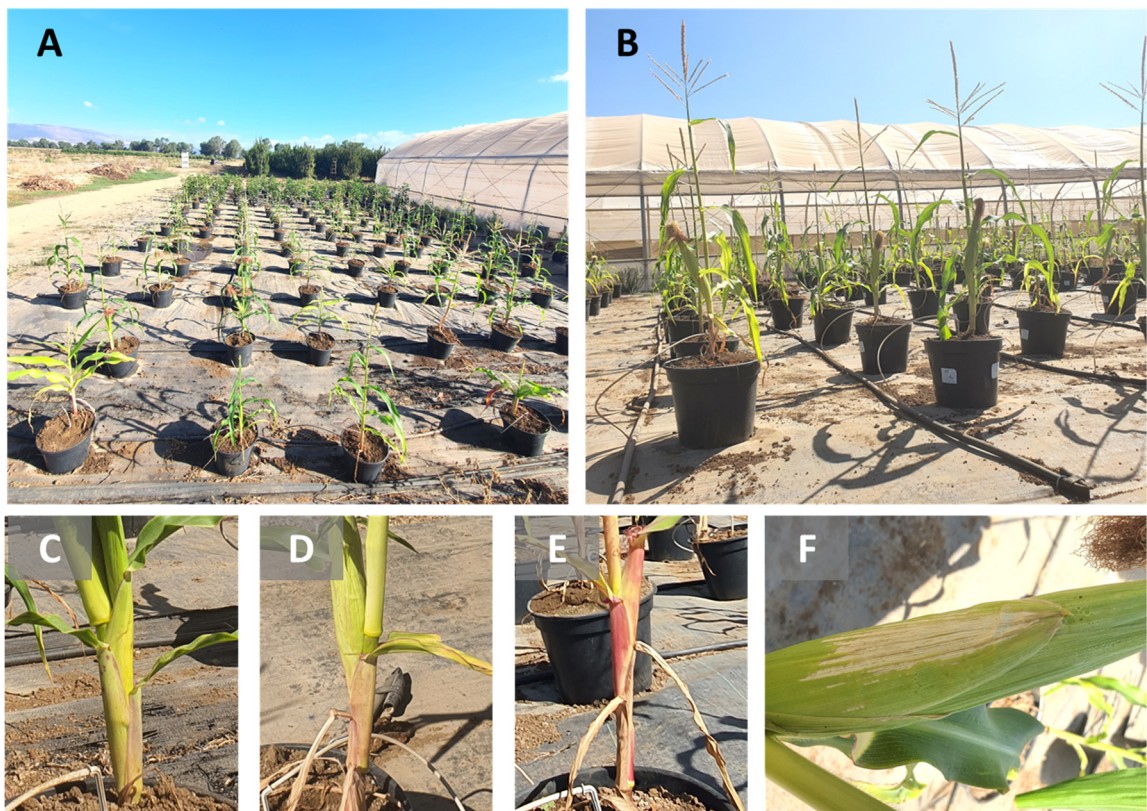

**Figure 6.** The open-enclosure full-season pots trial design and disease outbreak at harvest (78 days from sowing, 28 days from fertilization). This trial was carried out at the North R&D plantation farm (Hula Valley, Upper Galilee, northern Israel) with the Prelude cultivar (a late wilt disease susceptible sweet maize). The four *Trichoderma* species (Table 1), alone or combined with chemical protection, were tested against control plants (without biological and/or chemical treatments) in *M. maydis*-enriched soil. Azoxystrobin (5 mL of the Amistar S.C. preparation, which includes 1.25 mg of the active ingredient per pot) was applied at three intervals from the seeding: after 15, 30 and 44 days. A photo of the full-season trial is presented in (**A**). In the biological treatments, the plants developed well and reached fully matured cobs (**B**), whereas many of the unprotected control plants were diseased and grew poorly. In addition to the vitality of the overall plants and the dry leaves, the disease symptoms can be seen in the lower part of the stalk ((**C–E**), with increased severity) and the cobs' spathes (the large bracts surrounding the cobs, (**F**)).

At the above-surface peek evaluation made nine days post-sowing, some *Trichoderma* or *Trichoderma*-fungicide combinations had significant ($p < 0.05$) low emergence values. (Figure 7A). These treatments included *Trichoderma* sp. O.Y. 14707 (T14707) and *T. asperelloides* (T203), without or with Azoxystrobin, and *T. asperellum* (P1) with the fungicide. In contrast, at the end of the sprouting phase (day 40), this negative impact was meaningless (Figure 7B–F). On the contrary, biological treatments, except for the T7407, led to higher growth parameters and lower percentages of dry leaves. *T. asperelloides* (T203), in particular, led to profound ($p < 0.05$) growth promotion. This beneficial outcome was expressed in increased (compared to the non-protected control) shoot weight (165%), plant height (28%), number of plants with male flowers (189%) and lower (32%) number of dry leaves. Dry leaves present typical LWD symptoms: they change their color to light silver and yellow-brown and roll inward from the edges. The addition of chemical treatment didn't alter the impact of the bio-treatments, and statistically similar results were obtained with or without Azoxystrobin.

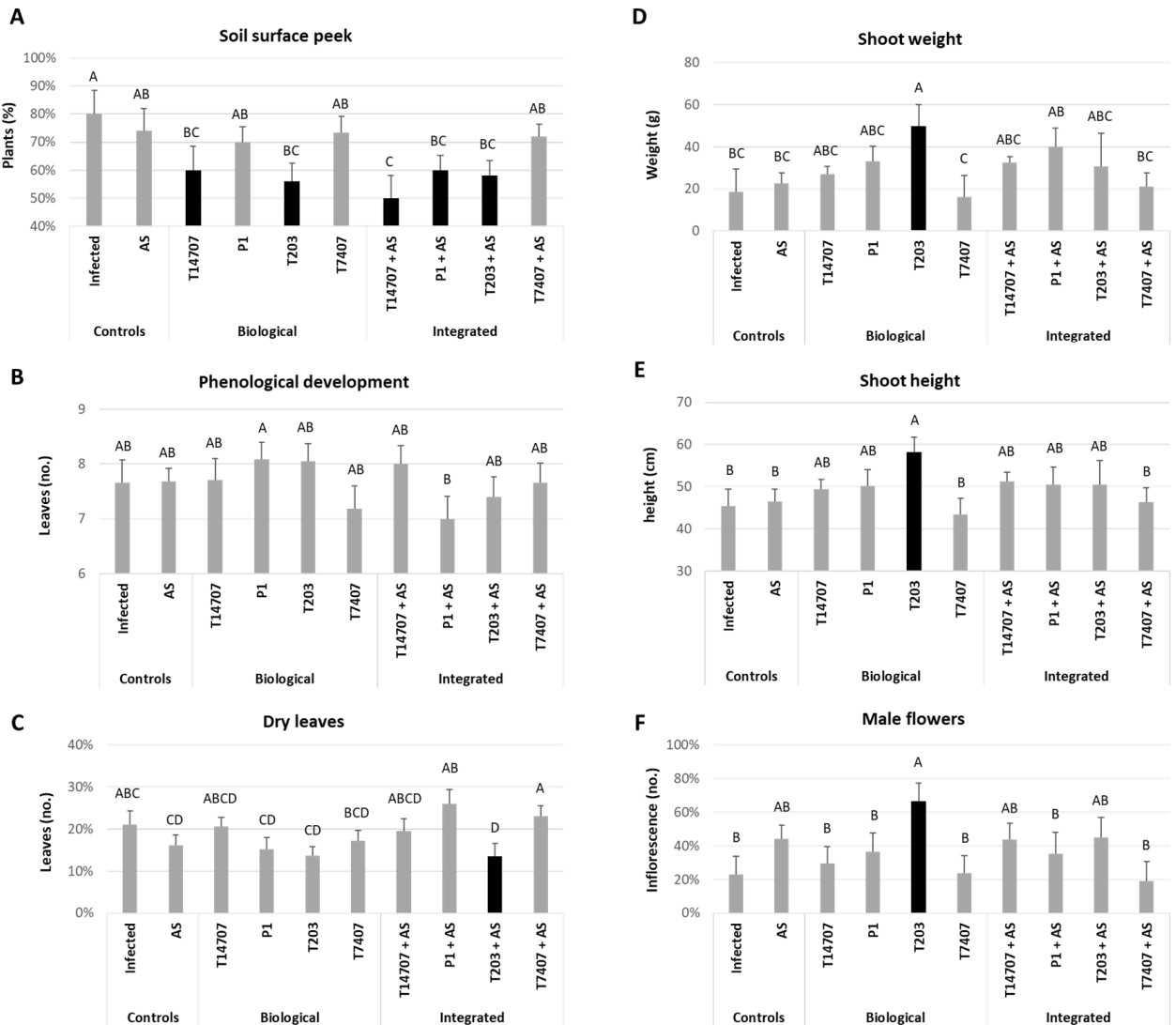

**Figure 7.** The open-enclosure trial above soil surface peek (emergence, day 9) and midseason sampling (day 40) growth evaluation. The experiment is described in Figure 6. The estimation included aboveground emergence percentages (**A**), phenological development of the plants (total number of leaves) (**B**), percentage of dry leaves (**C**), fresh weight of the shoot (**D**) and height (**E**), and percentage of plants with male flowers (**F**). The experiment was conducted in 16–26 repetitions (plants) per treatment. Error bars indicate standard error. Different letters above the chart's bars (A–D) represent an ANOVA test significant difference ($p < 0.05$).

Sixty-one days after sowing, the advantages of the biological control treatments were evident (Figure 8). The T203 treatment (without Azoxystrobin) achieved the best performance in promoting the height of the shoot (26%) under late wilt stress. Still, another *Trichoderma* application, *T. asperellum* (P1), excelled in improving the following growth parameters: number of cobs (100%), shoot height (18%), and in preventing the leaves from wilting (53% less dry leaves). Moreover, all other bio-shield treatments showed noticeable improvement compared to the unshielded control plants (8–17% higher shoot weight and 67–83% more cobs). Surprisingly, the greatest increase in number of leaves (21%, $p < 0.05$) was recorded in the T7407 (*T. longibrachiatum*) treatment. This parameter was unexpected since this treatment was the least beneficial; all the other measures had the lowest scores.

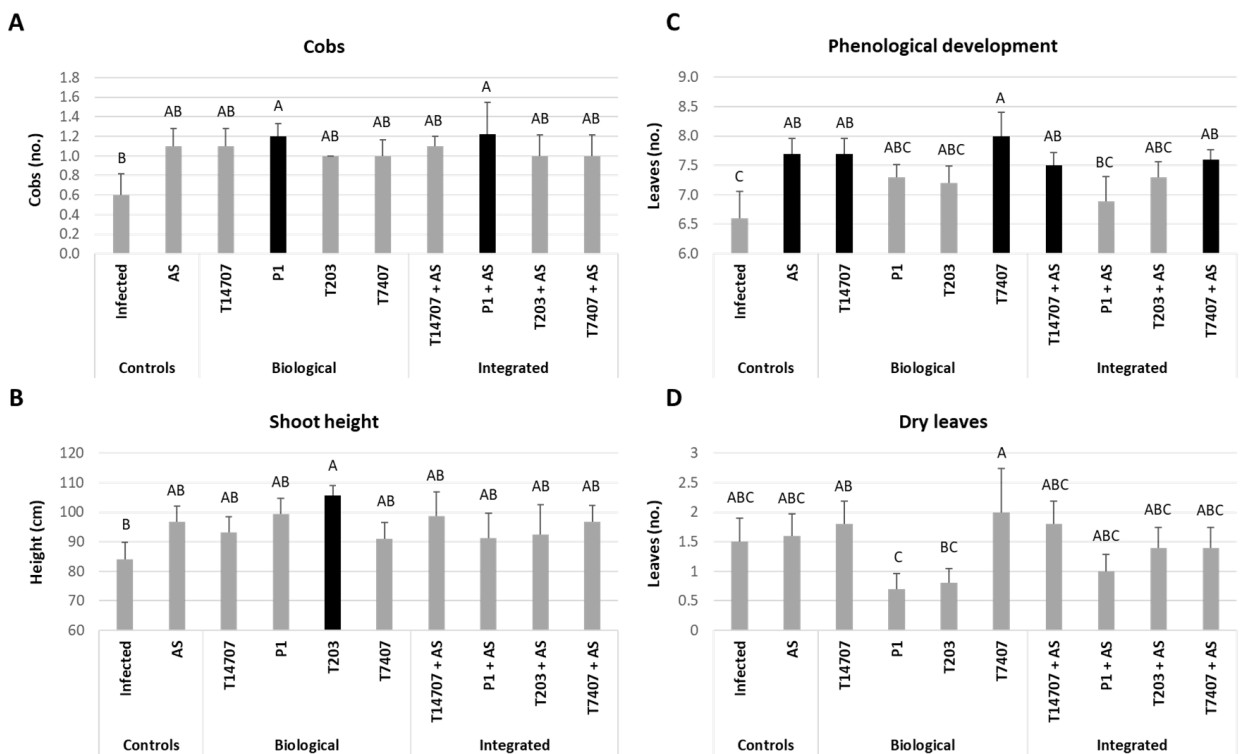

**Figure 8.** The open-enclosure trial growth parameters at day 61 from sowing. The experiment is described in Figure 6. The estimation included number of cobs (**A**), shoot height of the plants (**B**), phenological development (total number of leaves) (**C**) and number of dry leaves (**D**). Each value is a mean of 9–10 repetitions (plants per treatment). Error bars indicate standard error. Different letters above the chart's bars (A–C) represent an ANOVA test significant difference ($p < 0.05$).

At harvest (day 78 from sowing), both biological treatments, the T203 and the P1, maintained superiority in protecting the plant from late wilt disease and in accelerating growth compared to the unprotected control (Figure 9). All treatments improved cob yield (22–62%), but the most significant growth improvement was in the P1 soil enrichment (80%, $p < 0.05$). Remarkably, adding Azoxystrobin to the P1 resulted in a sharp loss of efficiency in this parameter and only a 7% improvement compared to the control. Such an impact was not evident in the wet shoot biomass in which the T203 and P1 showed significant enhancement ($p < 0.05$). Compared to the sole application of those treatments (71% and 76% shoot weight elevation, respectively), the integrated bio-chemo applications were also effective (51% and 80%, respectively). All other treatments had encouraging results (48–66% improvement in aboveground parts weight) that were similar to the sole Azoxystrobin treatment (57%).

Finally, the disease symptoms (Figures 9C and 10) and *M. maydis* infection level (pathogen's DNA within the roots, Figure 11) were closely tracked. Here, too, the sole P1 bio-soil enrichment stood out and contributed greatly to late wilt disease control. This impact was recorded in the all-plant dehydration assessment (29% more healthy plants). The other treatments (excluding the T7407) were also effective (without statistical significance) and led to 10–21% health recovery. For comparison, Azoxystroin when applied alone led to 14% health improvement.

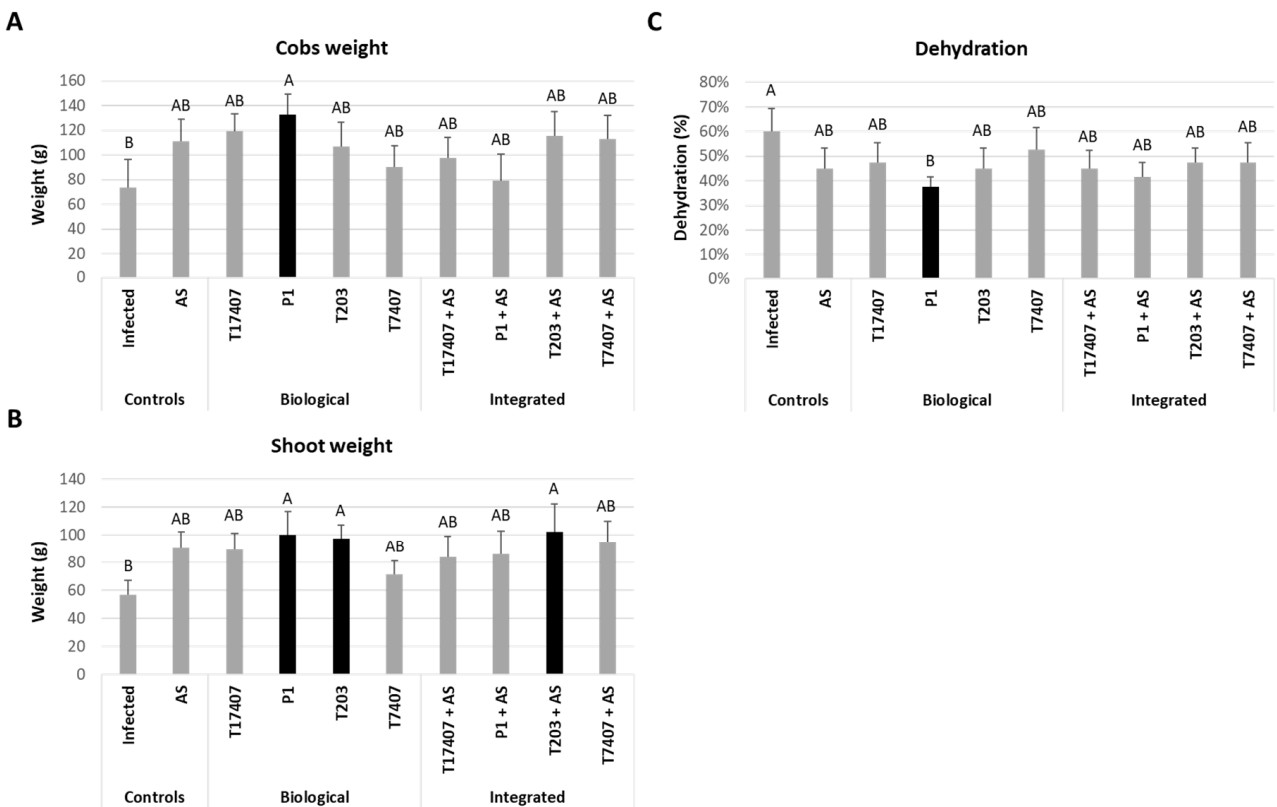

**Figure 9.** The open-enclosure trial growth parameters at harvest (day 78). The experiment is described in Figure 6. The estimation included fresh weight of the cobs (**A**), shoot wet biomass of the plants (**B**) and total dehydration (wilting) assessment according to four categories: 1—healthy, 2—symptoms, 3—diseased and 4—dead (**C**). Each value is a mean of 9–10 repetitions (plants per treatment). Error bars indicate standard error. Different letters above the chart's bars (A,B) represent an ANOVA test significant difference ($p < 0.05$).

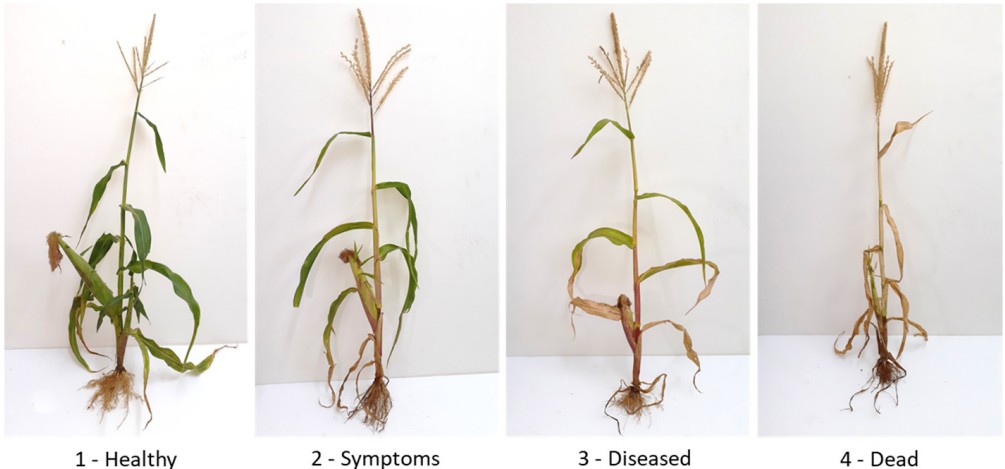

**Figure 10.** Wilting estimation scale. This scale was used to calculate the dehydration percentages at the different treatments at the end of season in the open-enclosure trial (day 78, Figure 9C). The plants' total health status was evaluated based on four categories: 1—healthy, 2—symptoms, 3—diseased and 4—dead.

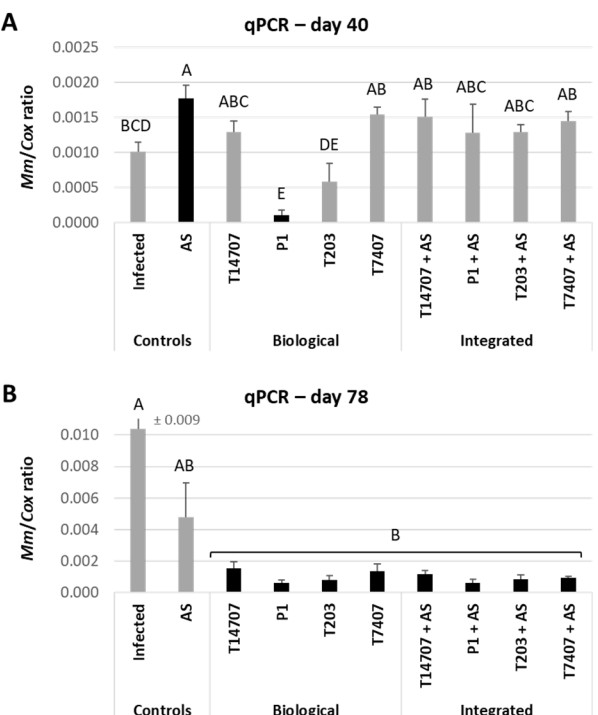

**Figure 11.** Real-time PCR-based monitoring of the relative amount of *M. maydis* DNA inside the plants' roots. This evaluation was made in the open-enclosure trial at midseason (**A**, day 40) and harvest (**B**, day 78). The relative amount of *M. maydis* DNA (*Mm*) normalized to the cytochrome C oxidase DNA (*Cox*) in the plants' roots is presented in the Y axis. Each value is a mean of 9–10 repetitions (plants per treatment). Error bars indicate standard error. Different letters (A–E) above the chart's bars represent an ANOVA test significant difference ($p < 0.05$).

Tracking the *M. maydis* pathogen in the roots of the host plants at the end of the sprouting stage (day 40) supported the growth indices variation and symptoms estimation (Figure 11A). Still, it revealed critical new information at the end of the season (growth day 78, Figure 11B). At midseason, the pathogen's DNA levels dropped by 90% ($p < 0.05$) and 42% in the P1 and T203 bio-protective treatments, respectively. Such a positive impact wasn't reached in the other treatments. Moreover, the addition of Azoxystrobin to the P1- and T203-treated plants abolished their protection against the *M. maydis* roots' establishment. Unexpectedly, the midseason *M. maydis* DNA levels in the sole Azoxystrobin treatment were significantly higher than the control. Such a result may be the consequence of several yet-to-be-explored reasons. These may include weakening the sprouts as a response to the fungicide phytotoxicity. Another explanation is the imbalance of the internally protective endophytes community that facilitates pathogen spread [43]. The true potential of the bio-shielding and bio-chemo protection was uncovered at harvest. Whereas the Azoxystrobin drip line irrigation alone was able to provide some (54%, insignificant) protection, all other treatments drastically and significantly ($p < 0.05$) reduced (85–94%) pathogen root infection levels. Among those successful applications, P1 and T203 maintained their superiority (94% and 92% pathogen suppression with or without Azoxystrobin addition, respectively).

## 4. Discussion

The current study is part of a continuous scientific effort to advance our understanding of maize late wilt disease and develop new ways to control it. The disease is considered the most severe maize disease in Israel [6,35] and Egypt [66], and a major concern in India [67], Spain [36] and Portugal [37]. It has been reported so far in 10 countries, but its prevalence and damage are predicted to expand due to global warming [18,68]. Over the last few decades, encouraging progress in our fundamental understanding of the disease and its

causal agent, the fungus *Magnaporthiopsis maydis*, opened the door for novel bio-friendly approaches to restrict the disease while maintaining high yields (for example, [48]). Such research challenges are now our top priority.

Integrated pest management was defined as the best mix of plant disease control strategies, taking into account crop yield, profit and safety profile [69]. Here, we studied the combination of a *Trichoderma*-based biological treatment and a well-established Azoxystrobin-based intervention to reduce the disease impact on a highly susceptible maize cultivar. This approach aimed at reducing the chemical effect on the environment, its potential health risks, and the possibility of fungicide resistance. All these advantages are enabled since the *Trichoderma* species can effectively attack and kill *M. maydis* in the soil and within the plant tissues [50], and develop throughout the season (thus maintaining high efficiency) [51]. According to other cases, these bio-pesticides can change with the pathogen in an evolutionary fashion (thus reducing resistance) [70–73]. The chemical treatment is still needed (in minimal dosages) in severe disease cases (heavily infested areas planted with susceptible genotypes) to weaken the pathogen and facilitate and enhance the bio-agents treatment. Indeed, the dosage of Azoxystrobin used in the current study was minimal, and neither the growth room sprouts assay nor the full-season trial could provide sufficient protection to the plants against the late wilt disease.

Several conclusions can be drawn from interpreting the results of this study. Most important, the *T. asperelloides* (T203) and *T. asperellum* (P1) biological treatments as sole treatments worked well and provided solid protection from the initial emergence of soil surface sprouts to maturation of the cobs. Figure 8 showed that the P1 treatment had the best plant growth promotion trait, which could be related to pathogen load (Figure 10). Indeed, this *Trichoderma* species is an endophyte isolated from maize grains [43] with a powerful secreted metabolite (6-Pentyl-$\alpha$-Pyrone) that inhibits the *M. maydis* pathogen at all stages of maize growth [59,74].

Combining biological and chemical treatments didn't disrupt bio-agent protection; in most cases, statistically similar results were obtained with or without the fungicide addition. This finding may be attributed to the high efficiency of the biological treatments that practically eliminate the disease. Yet, while bio-agents can substitute traditional fungicides, their disease management capacity is often incomplete and relies on uncontrollable environmental conditions [55]. So, the combined bio-chemo shielding tested here should be put to a more challenging trial. Such an examination could be carried out in a commercial field having a long record of strong disease outbreaks.

The results also presented two unpredicted and interesting findings. The *T. longibrachiatum* (T7404) that had previously exhibited high efficiency against the late wilt pathogen [51] had a relatively low impact here. In contrast, the control treatment with *Trichoderma* sp. O.Y. 14707 (T14707), which assumedly had no biocontrol activity versus *M. maydis* [50], successfully promoted plant growth and decreased disease symptoms. These results are likely the outcome of variations in the environmental conditions and the experiment's setting. From our experience, even highly *M. maydis*-infested cornfields that suffer from repeated disease outbreaks over several years had good years when no disease symptoms were recorded, and a normal yield was collected (Ofir Degani personal communication). As is well known in the phytopathology disease triangle, the environment, the host susceptibility (which also results from its health and developmental status) and the pathogen can each dramatically alter the pathogenesis outcome [75].

Semi-field trials simulate the situation in the field because they are done in the open air with natural field soil infected with local pathogen isolates and use common cultivars. They enable controlling inoculation load and water regime. They also prevent the treatments from affecting one another (even in a complete randomized dispersal of the pots). Controlling the water regime is particularly important since late wilt disease symptoms are enhanced under drought conditions and are moderate under optimal irrigation (high water potential) [7,24–27]. Despite the advantage of using pots in a field simulation, such trials cannot fully predict the field situation.

For instance, it was recently reported that adding *T. asperellum* with the seeding protects 42-day-old sprouts in a growth room (two-fold reduced pathogen root infection and growth promotion) [59]. The same procedure was less effective in a commercial field regarding growth and yield. Still, it decreased the cobs' symptoms by 11% and resulted in a nine-fold lower *M. maydis* DNA in the stem [59]. Here, the same *T. asperellum* treatment in an open-enclosure, whole-season pots trial led to remarkable growth recovery and disease repression. Thus, it can be inferred from this accumulated information that each control method should be tested for several seasons under different conditions to reveal its true full potential.

A combined biological and chemical approach for field crop protection is producing encouraging results. This approach was already tested in various plants against specific phytoparasitic fungi (reviewed in [55]). In particular, the application of *Trichoderma* spp. against soil-borne pathogens has gained considerable attention. For example, the combination of *T. virens* and thiophanate-methyl was more effective than either treatment alone against *Fusarium oxysporum* and *Fusarium solani* in a field assay in beans [76]. This combination significantly improved the plants' vegetative growth and yield. Likewise, the integrated application of *Trichoderma atroviride* Miller and *Trichoderma virens* Giddens with a low dose of fluazinam was discovered to be more efficient in controlling *Rosellinia necatrix* avocado's white rot than either treatment alone [77].

In corn, the spray application of difenoconazole-propiconazole followed by *Trichoderma harzianum* SH2303 was as successful in reducing *Cochliobolus heterostrophus* southern corn leaf blight as a sequential fungicide spraying. At the same time, the *Trichoderma* treatment alone was ineffective [56]. The integrated bio-chemo approach, therefore, allows a two-fold reduction in the fungicide's dose to control the disease. Another example that demonstrates this is the alteration of *T. harzianum* with dicarboximide fungicides against *Botrytis cinerea* gray mold on tomato plants. This treatment reduces (to half) the number of fungicide sprays [78].

Thus, a mixture of *Trichoderma* spp. with fungicides may be highly effective. Nevertheless, each combination should be examined carefully before being tested in plants to ensure the safety of the bio-agents [79]. To demonstrate this, the selection of iprodione-resistant *Trichoderma* spp. isolates is required for a combination of *T. viride* and the fungicide iprodione against *S. sclerotiorum* [80]. After that, the soil application of the iprodione-*Trichoderma* combination resulted in a synergistic impact that protected the cucumber plants.

The results obtained in the current study are the first step in developing a combined biological-chemical integrated control interphase that will be able to restrain late wilt disease damages even in high-risk areas. There is still much to learn before such a protocol could be implemented in commercial fields. Yet, the growing need to find green solutions for the control of filamentous fungi diseases is accelerating such studies. Hopefully, our increasing scientific database will allow presenting such a solution in the near future.

## 5. Conclusions

Integrated phytopathogen management strategies provide a reasonable compromise, considering both the desirability of biological control and the need for certain chemical control. *Trichoderma* species are particularly appropriate for this task because of their well-known biological control mechanism and wide use in agricultural applications [81]. Combining selected *Trichoderma* spp. with Azoxystrobin at a minimal dosage was presented here to overpower one of the most severe maize diseases in Israel, late wilt disease. The experiments included a sprouts pathogenicity test under controlled conditions and a semi-field, open-enclosure full-season pots trial. The results proved the biological shielding efficiency. The *T. asperelloides* (T203) and *T. asperellum* (P1) soil enrichment in particular provided significant ($p < 0.05$) protection against the pathogen *Magnaporthiopsis maydis* throughout the season. This reinforcement affected the growth indices and caused a drastic decrease in the fungal tissue establishment. This achievement was done without fungicide addition. Adding Azoxystrobin to those treatments resulted in similar (statistically equal) results. This work is the first to examine such a bio-chemo control interphase against late

wilt disease. The full potential of this approach should be put to the test under acute disease stress. Still, the fact that Azoxystrobin is harmless to the beneficial *Trichoderma* species over a complete semi-field condition is an excellent opening stage for subsequent future field studies.

**Author Contributions:** Conceptualization, A.G., B.H., E.D. and O.D.; data curation, A.G., B.H., E.D. and O.D.; formal analysis, A.G., B.H., E.D. and O.D.; funding acquisition, O.D; investigation, A.G., B.H., E.D. and O.D.; methodology, A.G., E.D. and O.D.; project administration, E.D. and O.D.; resources, O.D.; supervision, O.D.; validation, A.G., E.D. and O.D.; visualization, A.G., B.H., E.D. and O.D.; writing (original draft), O.D.; writing (review and editing), A.G., B.H., E.D. and O.D. All authors have read and agreed to the published version of the manuscript.

**Funding:** This work was supported by a one-year research grant (2022) from the Israel Field Crops Cultivation (FALCHA) Workers' Organization.

**Institutional Review Board Statement:** Not applicable.

**Informed Consent Statement:** Not applicable.

**Data Availability Statement:** The datasets generated and/or analyzed during the current study are available from the corresponding author upon reasonable request.

**Acknowledgments:** We would like to thank Onn Rabinovitz (Migal—Galilee Research Institute, Israel) for his many helpful suggestions and Galia Shofman (Migal—Galilee Research Institute and Tel-Hai College, Israel) for her technical assistance and wise advice. We would also like to thank Menashe Levi (MIGAL—Galilee Research Institute, Israel, R&D North), who assisted in conducting the pot semi-field experiment.

**Conflicts of Interest:** The authors declare no conflict of interest.

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
