# Peer review of "Integrated Biological and Chemical Control against the Maize Late Wilt Agent Magnaporthiopsis maydis"

_soilsystems, doi:10.3390/soilsystems7010001_

Round 1

Reviewer 1 Report

The manuscript described a control method for late wilt of maize by using chemical fungicide together with metabolize of bio control agents. Generally, it's well designed and presented. And the results will give a case for related research and field application. 

Some minor points:

1. Is the pathogen stain isolated in local feilds? If yes, please statement. If not, should prove its resembles to the local ones. And, if more than one strain used, better.

2. In the whole manuscript, no images was presented for pathogen, biocontrol fungus, on petri dishes or under microscope. It is necessary to add some.

3. The manuscript will be  better after a writing improvement.

Author Response

Responses to Reviewer 1’s comments

We thank the reviewer for investing substantial efforts, which undoubtedly contribute to this manuscript. The remarks and suggestions improved this paper’s scientific soundness and accuracy. Your contribution is greatly appreciated.

The manuscript described a control method for the late wilt of maize by using chemical fungicide together with metabolize of bio-control agents. Generally, it’s well-designed and presented. And the results will give a case for related research and field application. 

Reply: Thank you for the positive evaluation of our manuscript. All your remarks and suggestions were addressed carefully and thoroughly, as detailed below.

Some minor points:

  1. Is the pathogen strain isolated in local fields? If yes, please state that. If not, should prove its resembles to the local ones. And, if more than one strain used, better.

Reply: Thank you for this important remark. Yes, the pathogen strain was isolated from a local field. This isolate was formerly recovered in 2001 from a Kibbutz Sde Nehemia cornfield in the Hula Valley in Upper Galilee in northern Israel from Jubilee cv. maize plants that showed dehydration symptoms. It was lately evaluated and compared to other 16 M. maydis strains in our fungal library, collected from maize fields across Israel, and proved to be a mild aggressive – virulence rank 7 out of 17 strains (see Shofman et al., (2022), Fungal Biology, 126(11-12), 793-808, https://doi.org/10.1016/j.funbio.2022.10.003.

The following explanation was added regarding this isolate to the text (lines 126-131):

“It was formerly recovered in 2001 from a Kibbutz Sde Nehemia cornfield in the Hula Valley (Upper Galilee, northern Israel) from Jubilee cv. maize plants that showed dehydration symptoms. It was recently evaluated and compared to 16 other M. maydis strains in our fungal library collected from maize fields across Israel, and proved to be a mild aggressive – virulence rank 7 out of 17 strains [35].”

  1. In the whole manuscript, no images were presented for pathogen, biocontrol fungus, on Petri dishes or under a microscope. It is necessary to add some.

Reply: As suggested, a new figure, Figure 1, was added. The figure shows a mycoparasitism assay of the Trichoderma spp. used in this study against the maize late wilt pathogen M. maydis. The text was updated accordingly.

  1. The manuscript will be better after a writing improvement.

Reply: A professional English scientific copy editor edited the entire manuscript. The manuscript was carefully and thoroughly re-checked after the revision. We tried our best to simplify the writing and make the text more coherent and clearer.

Reviewer 2 Report

The authors report an integrated strategy for control of fungal disease in maize. The integrated strategy is based on combination of Trichoderma species + Azoxystrobin. The authors measured plant growth parameters, disease symtoms, and pathogen load. Based on results at 78 days, the better result was provided by P1-treatment (T. asperellum P1), since i) plant growth promotion, ii) symptom reduction and iii) pathogen load reduction were observed.  In other words, the treatments (biological and integrated) showed differential results.

My Comments:

1)     I recommend to authors discuss about utility of integrated strategy (biological + chemical), considering that aplication of P1 revealed aceptable results for control of maize fungal disease.

2)     Extensive editing of English language and style is required. Example the use of apostrophes. There are many long sentences and complex paragraphs, that are not easily to follow up. I recommend the use of short and consise sentences.

3)     Figures look blurry, increasing dpi for a better resolution. Some figures can be employed as supplementary figures.

4)     Some figures were not properly explained, instead only mentioned.

5)     The figure 10ª revealed that pathogen load is greater in AS-treatment that in infected-treatment, authors should clarify the discrepancy.

6)     I strongly recommend to authors focus on results whose showed statistical significance difference, with the aim to summarize information and easily follow up the paper.

Below some specific comments:

ABSTRACT. The authors should omit values and focus in concise advances reached.

Line 39, 47 and many others: season end, plant roots… Comment: omit apostrophes.

Line 48: “…and establishes a vascular disease”.

Line 59: what means LWD?

Line 73: “…can protect the plants from…”

Line 66, 68 and many others: “…tested over the years [37]. Some of…” Only indicate reference and omit additional text.

Line 95: “Because in severe disease cases, it may not be sufficient”. Authors should mention likely reason for this situation.

INTRODUCTION. Introduction section is too long, I suggest to authors rewrite it in concise way.

MATERIAL AND METHODS.

Line 115: “To reduce Azoxystrobin to a minimum, we ran a plate assay with different preparation concentrations to set the minimal inhibition concentration (MIC) needed to control Magnaporthiopsis maydis, causal agent of maize late wilt (LWD)”. Comment: Indicate concentrations of Azo, # replicates, culture media.

Line 164: indicate number of employed sedes/replicate

Line 283: Information about primers should be presented as table: indicate name primer, sequence, amplicon size, PCR cycling. Authors should indicate if primers were designed or respective reference.

Line 288: “…according to 2-ΔΔCt method”.

RESULTS

Line 298 and many other lines: “…an integrated control strategy against M. maydis, agent casual of maize LWD….” Authors should use resumed scientific name of pathogen and plant disease.

Line 299: “…integrated control strategy based on…”

Line 304: “…with the biological agents.

Line 306 and others: The results should be written in past tense.

Figure 1, 2, 3, 4, 5, 6, 7, 8, 9 10 look blurry, increasing dpi for a better resolution.

Line 373: “Tracking the pathogen’s DNA within the host plants’ roots revealed the integrated bio-chemo control method potential”. This sentence belongs to conclusion section, since results indicated a differential decrease of pathogen load. Authors would discuss in quantitative terms.

Line 391: “…such growth allowed us to maintain control of the infection load, soil composition,…”

Line 392-397: The figure 5 was not properly explained.

Line 395: “As expected…”

Line 399-410: The paragraph is very important since corresponds to figure 6, performance of integrated strategy at midseason (40 days). However, the paragraph contains long sentences. The paragraph is not easily Comprehensible. I suggest to authors rewrite it, using short and consise sentences.

Line 416: “…against unshielded control plants in M. maydis-enriched soil…” My comment: what means “unshielded control plants”?

Figure 5: The disease symptoms were related to disease categories? Authors should indicate the categorie in respective figure.

Figure 6: What means “soil Surface peek”?

Figure 6 and 7: What means “dry leaves”? Does it correspond to fungal spot in leaves? The authors would clarify the measured parameters in respective section.

Line 404: “…(Figure 6B-F). On the other hand, biological treatments…

Line 437: “…bio-shield treatments…” Authors would homogenize and define terms in the paper, at respective section. Example: bio-shield ßàbiological treatments?

Line 440: “…his parameter was unexpected since this treatment was the less beneficial”. My Comment: I recommend to authors focus on results whose showed statistical significance difference.

Figure 8: Dehidratation corresponds to plant wilting? The authors would clarify the measured parameters in respective section.

Figures 8C, 9 and 10 were not properly explained. The units of figure 10 were unclear. The authors would clarify if the quatification of pathogen was relative or absolute.

The figure 10ª revealed that pathogen load is greater in AS-treatment that in infected-treatment, authors should clarify the discrepancy.

DISCUSSION.

Lines 539: The figure 8 showed that P1-treatment had the best plant growth promotion trait which could be related to pathogen load (figure 10). The authors should discuss such results.

Line 551: “…in the open air…” Comment: does it refer to mimic “intensive production unit”?

CONCLUSION

Line 600: “Integrated phytopathogen management strategies provide…”

Line 611: “…fungal tissue establishment. This…”

Author Response

Responses to Reviewer 2’s comments

We would like to express our sincere appreciation to the reviewer for the essential and helpful advice. The time and effort invested are greatly appreciated and certainly contributed to the manuscript and improved it. Thank you.

The authors report an integrated strategy for control of fungal disease in maize. The integrated strategy is based on combination of Trichoderma species + Azoxystrobin. The authors measured plant growth parameters, disease symtoms, and pathogen load. Based on results at 78 days, the better result was provided by P1-treatment (T. asperellum P1), since i) plant growth promotion, ii) symptom reduction and iii) pathogen load reduction were observed. In other words, the treatments (biological and integrated) showed differential results.

Reply: Thank you for the evaluation of our manuscript. All your remarks and suggestions were addressed carefully and thoroughly, as detailed below.

My Comments:

1) I recommend to authors discuss about utility of integrated strategy (biological + chemical), considering that aplication of P1 revealed aceptable results for control of maize fungal disease.

Reply: This is an important suggestion. Thank you. We addressed this aspect as follows:

  • Abstract; lines 12-14: “On the other hand, biocontrol agents such as Trichoderma provide a green solution but often cannot shield the plants from aggressive disease outbreaks.”
  • Introduction; lines 84-98: “Thus, why not use this biological approach exclusively? Because in severe disease cases, it may not be sufficient. Although Trichoderma-based biological control is widely studied for its eco-protection advantage, its application is often challenged by natural stress in farming, leading to unpredictable control effects [52-54]. A proposed new control strategy maintains high chemical effectiveness while drastically reducing the dosages applied by combining both chemical and biological control methods [55]. It was previously reported that the synergistic application of low-toxic chemical fungicides and biocontrol agents could improve biocontrol stability and efficiency against plant diseases, ultimately reducing the use of chemical fungicides. An example is the combined application of Trichoderma harzianum (strain SH2303) and difenoconazole-propiconazole in controlling maize southern corn leaf blight disease caused by Cochliobolus heterostrophus [56]. Such an environmentally-friendly solution might also prevent the increasing fungicide resistance problem (reducing the selection pressure on pathogens caused by pesticide overuse, thereby, the chances of resistance evolving).”
  • Discussion; lines 542-547: “The chemical treatment is still needed (in minimal dosages) in severe disease cases (heavily infested areas planted with susceptible genotypes) to weaken the pathogen and facilitate and enhance the bio-agents treatment. Indeed, the dosage of Azoxystrobin used in the current study was minimal, and neither the growth room sprouts assay nor the full-season trial could provide sufficient protection to the plants against the late wilt disease.”
  • Lines 556-563: “Combining biological and chemical treatments didn’t disrupt bio-agent protection; in most cases, statistically similar results were obtained with or without the fungicide addition. This finding may be attributed to the high efficiency of the biological treatments that practically eliminate the disease. Yet, while bio-agents can substitute traditional fungicides, their disease management capacity is often incomplete and relies on uncontrollable environmental conditions [55]. So, the combined bio-chemo shielding tested here should be put to a more challenging trial. Such an examination could be carried out in a commercial field having a long record of strong disease outbreaks.”
  • Conclusion; lines 637-642: “Adding Azoxystrobin to those treatments resulted in similar (statistically equal) results. This work is the first to examine such a bio-chemo control interphase against late wilt disease. The full potential of this approach should be put to the test under acute disease stress. Still, the fact that Azoxystrobin is harmless to the beneficial Trichoderma species over a complete semi-field condition is an excellent opening stage for subsequent future field studies.”

2) Extensive editing of English language and style is required. Example the use of apostrophes. There are many long sentences and complex paragraphs, that are not easily to follow up. I recommend the use of short and consise sentences.

Reply: A professional English scientific copy editor edited the entire manuscript. The manuscript was carefully and thoroughly re-checked after the revision. We tried our best to simplify the writing and make the text more coherent and clearer.

3) Figures look blurry, increasing dpi for a better resolution. Some figures can be employed as supplementary figures.

Reply: All original files of the figures are at a high resolution (exceeding 300 dpi). The figures embedded in the manuscript may be kept at a lower resolution to keep the file size reasonable. We will submit the original figures to the journal.

4) Some figures were not properly explained, instead only mentioned.

Reply: All of the figure legends were checked thoroughly and carefully corrected to achieve better clarity and coherence.

5) Figure 10 revealed that pathogen load is greater in AS-treatment that in infected-treatment, authors should clarify the discrepancy.

Reply: We explained this issue in our response to the specific comments below.

6) I strongly recommend to authors focus on results whose showed statistical significance difference, with the aim to summarize information and easily follow up the paper.

Reply: you are right, and the description of the results should focus on those that gained significant differences from the control. So, we deleted the sentences that referred to non-statistically significant results. All remaining result descriptions concentrate on those that achieved a high difference (p < 0.05) compared to the control.

Below are some specific comments:

ABSTRACT.

The authors should omit values and focus in concise advances reached.

Reply: We deleted most of the values as suggested. The paragraph now reads (lines 23-26): “At harvest, the P1 and T203 bio-shielding exhibited the best parameters (statistically significant) in plant growth promotion, yield increase and late wilt protection (up to 29% health recovery and 94% pathogen suppression tracked by real-time PCR).” We believe it is important to present the health values, which are unique to late wilt disease.

INTRODUCTION.

Line 39, 47 and many others: season end, plant roots… Comment: omit apostrophes.

Reply: The apostrophes were omitted from these sentences and in other places according to the English scientific copy editor’s advice.

Line 48: “…and establishes a vascular disease”.

Reply: Corrected as advised.

Line 59: what means LWD?

Reply: LWD is an abbreviation for late wilt disease. We apologize for this editing mistake. This explanation was added to line 53 in the Introduction. It already existed the first time these abbreviations appeared in the Materials and Methods and Results sections.

Line 73: “…can protect the plants from…”

Reply: Corrected as advised.

Line 66, 68 and many others: “…tested over the years [37]. Some of…” Only indicate reference and omit additional text.

Reply: We corrected the sentence as advised by the reviewer and made changes to the text in many other places. Hopefully, this answers your concern.

Line 95: “Because in severe disease cases, it may not be sufficient”. Authors should mention likely reason for this situation.

Reply: This is an important remark; thank you. The following explanation was added to the Introduction (lines 86-95):

“Although Trichoderma-based biological control is widely studied for its eco-protection advantage, its application is often challenged by natural stress in farming, leading to unpredictable control effects [52-54]. A proposed new control strategy maintains high chemical effectiveness while drastically reducing the dosages applied by combining both chemical and biological control methods [55]. It was previously reported that the synergistic application of low-toxic chemical fungicides and biocontrol agents could improve biocontrol stability and efficiency against plant diseases, ultimately reducing the use of chemical fungicides. An example is the combined application of Trichoderma harzianum (strain SH2303) and difenoconazole-propiconazole in controlling maize southern corn leaf blight disease caused by Cochliobolus heterostrophus [56].”

The introduction section is too long, I suggest to authors rewrite it in concise way.

Reply: We carefully reorganized and edited the Introduction to make it more focused, short and straightforward. The following sentences or paragraphs were deleted from the Introduction chapter, according to your advice:

  • “The pathogen stays dormant in the soil [17], and, under favorable conditions (permeable host plant and appropriate temperature and humidity), infiltrates the plant roots and establishes a vascular disease [18].”
  • “In maize, this pathogenic initiation stage lasts about three weeks, at the end of which the fungus reaches the first internode above the ground [11]. This stage is crucial for disease development since after ca. 50 days, the plants may become immune to new infections, except if the roots were wounded [11].”
  • “Such a scenario already took place in Israel with the Royalty cultivar.”
  • “Green solutions to LWD are gaining success, and accumulated evidence of their positive potential is Despite this trend, many farmers do not apply them and prefer traditional chemical methods.”
  • “Since Israel is highly affected by LWD, it will be important to examine the integrated control method against Israeli maydis strains.”
  • Minor changes were made in other places to eliminate unnecessary words and make the text clearer and more coherent.

MATERIALS AND METHODS.

Line 115: “To reduce Azoxystrobin to a minimum, we ran a plate assay with different preparation concentrations to set the minimal inhibition concentration (MIC) needed to control Magnaporthiopsis maydis, causal agent of maize late wilt (LWD)”. Comment: Indicate concentrations of Azo, # replicates, culture media.

Reply: This information is presented in detail in Section 2.3 (Azoxystrobin inhibition plates assay). So, there is no need to present it in the opening section (2.1) that aimed at describing the study rationale of the design. Still, we added a reference to Section 2.3 to make this point clearer:

“To reduce Azoxystrobin to a minimum, we ran a plate assay (see Section 2.3) with different preparation concentrations and set the minimal inhibition concentration (MIC) point needed to control the maize late wilt (LWD) pathogen, Magnaporthiopsis maydis.”

Line 164: indicate number of employed sedes/replicate

Reply: The missing information was added to the text (lines 182-183): “Each group of seeds was tested in six repetitions. Each replication is a plate containing 10 seeds.”

Line 283: Information about primers should be presented as table: indicate name primer, sequence, amplicon size, PCR cycling. Authors should indicate if primers were designed or respective reference.

Reply: We agree. The primers information is now presented in a table (Table 2) and includes all the information suggested by the reviewer.

Line 288: “…according to 2-ΔΔCt method”.

Reply: We calculate the relative gene expression using the mean ΔCt value (threshold cycle). The method we used for calculating relative gene expression from the quantification cycle (Cq) values obtained by the qPCR analysis is well explained in the following citation: Haimes, J., and M. Kelley. “Demonstration of a ΔΔCq calculation method to compute relative gene expression from qPCR data.” Thermo Scientific Tech Note 1 (2010) (full-text link).

RESULTS

Line 298 and many other lines: “…an integrated control strategy against M. maydis, agent casual of maize LWD….” Authors should use resumed scientific name of pathogen and plant disease.

Reply: The full name of the pathogen and the disease were added. The sentence now reads: “This study examined, for the first time, an integrated control interphase against the maize late wilt disease (LWD) pathogen, Magnaporthiopsis maydis.”

Line 299: “…integrated control strategy based on…”

Reply: Corrected as advised.

Line 304: “…with the biological agents.”

Reply: Corrected as advised.

Line 306 and others: The results should be written in past tense.

Reply: The Results section was carefully checked, and the writing was corrected to past tense.

Figure 1, 2, 3, 4, 5, 6, 7, 8, 9 10 look blurry, increasing dpi for a better resolution.

Reply: as we explained above, all original files of the figures are at a high resolution (exceeding 300 dpi). The figures embedded in the manuscript may be kept at a lower resolution to keep the file size reasonable. We will submit the original figures to the journal.  

Line 373: “Tracking the pathogen’s DNA within the host plants’ roots revealed the integrated bio-chemo control method potential”. This sentence belongs to conclusion section, since results indicated a differential decrease of pathogen load. Authors would discuss in quantitative terms.

Reply: The sentence was deleted from the Results section.

Line 391: “…such growth allowed us to maintain control of the infection load, soil composition,…”

Reply: Corrected as advised.

Line 392-397: The figure 5 was not properly explained.

Reply: You are right. The Figure 5 (now Figure 6) legend was corrected and clarified.

Line 395: “As expected…”

Reply: Corrected as advised.

Line 399-410: The paragraph is very important since corresponds to figure 6, performance of integrated strategy at midseason (40 days). However, the paragraph contains long sentences. The paragraph is not easily Comprehensible. I suggest to authors rewrite it, using short and consise sentences.

Reply: Thank you. The entire paragraph was edited (especially long sentences were cut into two parts), making it easier to follow and understand.

The paragraph now reads (lines 416-429):

“At the above-surface peek evaluation made nine days post-sowing, some Trichoderma or Trichoderma-fungicide combinations had significant (p < 0.05) low emergence values. (Figure 7A). These treatments included Trichoderma sp. O.Y. 14707 (T14707), T. asperellum (P1), without or with Azoxystrobin, and T. longibrachiatum (T7404). In contrast, at the end of the sprouting phase (day 40), this negative impact was meaningless (Figure 7B-F). On the other hand, biological treatments, except for the T7407, led to higher growth parameters and lower percentages of dry leaves. T. asperelloides (T203), in particular, led to profound (p < 0.05) growth promotion. This beneficial outcome was expressed in increased (compared to the non-protected control) shoot weight (165%), plant height (28%), number of plants with male flowers (189%) and lower (32%) number of dry leaves. Dry leaves present typical LWD symptoms: they change their color to light silver and yellow-brown and roll inward from the edges. The addition of chemical treatment didn’t alter the impact of the bio-treatments, and statistically similar results were obtained with or without Azoxystrobin.”

Line 416: “…against unshielded control plants in M. maydis-enriched soil…” My comment: what means “unshielded control plants”?

Reply: We meant to say without biological and/or chemical treatments. The sentence was corrected and now reads (lines 434-436): “The four Trichoderma species (Table 1), alone or combined with chemical protection, were tested against control plants (without biological and/or chemical treatments) in M. maydis-enriched soil.”

Figure 5: The disease symptoms were related to disease categories? Authors should indicate the categorie in respective figure.

Reply: Most likely, yes. As stated in the Materials and Methods (lines 278-280), “At the end of the season, the total health status of the plants was evaluated based on four categories: 1 – healthy, 2 – symptoms, 3 – diseased and 4 – dead.” The wilting estimation scale photos are presented in Figure 10 (Figure 9 in the previous version). The lower stem symptoms reflect the whole plant health symptoms, especially in severely diseased plants, but were not used to set the plants’ health status in this work.

Figure 6: What means “soil Surface peek”?

Reply: Above the soil-surface peek percentages, i.e., the percentages above the soil-surface emergence. The sentence was corrected for better clarity: “The open-enclosure trial above soil surface peek (emergence, day 9) and midseason sampling (day 40) growth evaluation.” (lines 444-445).

Figure 6 and 7: What means “dry leaves”? Does it correspond to fungal spot in leaves? The authors would clarify the measured parameters in respective section.

Reply: The first symptom of late wilt disease is moderate plant wilting progressing upwards. The leaves change their color to light silver and yellow-brown and roll inward from the edges. Thus, by saying “in dry leaves,” we meant that the whole leaf was dried with a yellow-brown color. The following explanation was added to the text (lines 425-427):

“Dry leaves present typical LWD symptoms: they change their color to light silver and yellow-brown and roll inward from the edges.”

Line 404: “…(Figure 6B-F). On the other hand, biological treatments…

Reply: Corrected as advised.

Line 437: “…bio-shield treatments…” Authors would homogenize and define terms in the paper, at respective section. Example: bio-shield ßàbiological treatments?

Reply: Since these are synonyms and are commonly used in the scientific literature, we used them to enrich the text and avoid sounding monotonous. The English scientific copy editor checked and confirmed that all phrases were used correctly.  

Line 440: “…his parameter was unexpected since this treatment was the less beneficial”. My Comment: I recommend to authors focus on results whose showed statistical significance difference.

Reply: The above sentence referred to a statistically significant result. We have highlighted this fact in the text (lines 458-460): “Surprisingly, the greatest increase in number of leaves (21%, p < 0.05) was recorded in the T7407 (T. longibrachiatum) treatment. This parameter was unexpected since this treatment was the least beneficial; all the other measures had the lowest scores.”

Still, you are right, and the description of the results should focus on those that gained significant differences from the control. So, the following sentence was deleted: “ The same tendencies were recorded when Azoxystrobin was added with the biological pesticides (without a statistically significant difference between the two).”

Figure 8: Dehidratation corresponds to plant wilting? The authors would clarify the measured parameters in respective section.

Reply: Yes. Dehydration and wilting are synonyms that describe the disease outcome. We added a short explanation to Figure 9 (previously Figure 8) to clarify this issue.

Figures 8C, 9 and 10 were not properly explained. The units of figure 10 were unclear. The authors would clarify if the quatification of pathogen was relative or absolute.

Reply: We double-checked those figure legends as per your recommendation. The quantification of the pathogen is in relative DNA. This information was added to the caption of Figure 11 (previously Figure 10): “Figure 11. Real-time PCR-based monitoring of the relative amount of M. maydis DNA inside the plants’ roots.” The information was also described in the text below: “The relative amount of M. maydis DNA (Mm) normalized to the cytochrome C oxidase DNA (Cox) in the plants’ roots is presented in the Y axis.”

The figure 10ª revealed that pathogen load is greater in AS-treatment that in infected-treatment, authors should clarify the discrepancy.

Reply: This issue should indeed be clarified; thank you. The following paragraph was added to the text (lines 493-498): “Unexpectedly, the midseason M. maydis DNA levels in the sole Azoxystrobin treatment were significantly higher than the control. Such a result may be the consequence of several yet-to-be-explored reasons. These may include weakening the sprouts as a response to the fungicide phytotoxicity. Another explanation is the imbalance of the internally protective endophytes community that facilitates pathogen spread [43].”

DISCUSSION.

Lines 539: The figure 8 showed that P1-treatment had the best plant growth promotion trait which could be related to pathogen load (figure 10). The authors should discuss such results.

Reply: Indeed. The following explanation was added to the discussion (lines 551-555): “Figure 8 showed that the P1 treatment had the best plant growth promotion trait, which could be related to pathogen load (Figure 10). Indeed, this Trichoderma species is an endophyte isolated from maize grains [43] with a powerful secreted metabolite (6-Pentyl-α-Pyrone) that inhibits the M. maydis pathogen at all stages of maize growth [59,74].”

Line 551: “…in the open air…” Comment: does it refer to mimic “intensive production unit”?

Reply: No, the sentence simply states, “Semi-field trials simulate the situation in the field because they are done in the open air with natural field soil infected with local pathogen isolates and use common cultivars.” (lines 576-577). We changed the word “mimic” to “simulate” for better clarity.

CONCLUSION

Line 600: “Integrated phytopathogen management strategies provide…”

Reply: Corrected as advised.

Line 611: “…fungal tissue establishment. This…”

Reply: Corrected as advised.